# Simulation of an Elastic Rod Whirling Instabilities by Using the Lattice Boltzmann Method Combined with an Immersed Boundary Method

Suresh Alapati [1,*], Wooseong Che [1], Sunkara Srinivasa Rao [2] and Giang T. T. Phan [3,4,*]

1 Department of Mechatronics Engineering, Kyungsung University, 309 Suyeong-ro, Nam-gu, Busan 48434, Republic of Korea; wsche@ks.ac.kr
2 Department of Electronics and Communication Engineering, Koneru Lakshmaiah Education Foundation, Bowrampet, Hyderabad 500043, Telangana, India; srinivasarao.s@klh.edu.in
3 Institute of Fundamental and Applied Sciences, Duy Tan University, Ho Chi Minh City 700000, Vietnam
4 Faculty of Natural Sciences, Duy Tan University, Da Nang 550000, Vietnam
* Correspondence: suresh@ks.ac.kr or sureshalapatimech@gmail.com (S.A.); phantthuygiang@duytan.edu.vn (G.T.T.P.)

**Abstract:** Mathematical modeling and analysis of biologically inspired systems has been a fascinating research topic in recent years. In this work, we present the results obtained from the simulation of an elastic rod (that mimics a flagellum axoneme) rotational motion in a viscous fluid by using the lattice Boltzmann method (LBM) combined with an immersed boundary method (IBM). A finite element model consists of a set of beam and truss elements used to discretize the flagellum axoneme while the fluid flow is solved by the well-known LBM. The hydrodynamic coupling to maintain the no-slip boundary condition between the fluid and the elastic rod is conducted with the IBM. The rod is actuated with a torque applied at its base cross-section that acts as a driving motor of the axoneme. We simulated the rotational dynamics of the rod for three different rotational frequencies (low, medium, and high) of the motor. To compare with previous publication results, we chose the sperm number $S_p = L[(4\pi\mu\omega)/(EI)]^{1/4}$ as the validation parameter. We found that at the low rotational frequency, $f = 1.5$ Hz, the rod performs stable twirling motion after attaining an equilibrium state (the rod undergoes rigid rotation about its axis). At the medium frequency, $f = 2.65$ Hz, the rod undergoes whirling motion, where the tip of the rod rotates about the central rotational axis of the driving motor. When the frequency increases further, i.e., when it reaches the critical value, $f_c \approx 2.7$ Hz, the whirling motion becomes over-whirling, where the tip of the filament falls back to the base and performs a steady crank-shafting motion. All three rotational dynamics, twirling, whirling, and over-whirling, and the critical value of rotational frequency are in good agreement with the previously published results. We also observed that our present simulation technique is computationally more efficient than previous works.

**Keywords:** lattice Boltzmann method; immersed boundary method; flagellum mathematical analysis; axoneme mathematical model; twirling

**MSC:** 65K05; 65M75; 76M27; 76A02

## 1. Introduction

Bio-fluid dynamics is one of the most fascinating and emerging research fields since biological flows, such as blood flow through arteries, sperm, cilia, and bacterial motion in a viscous fluid, are ubiquitous. The propulsion of a bacterial flagellum (a curvy helical filament of protein) in a viscous fluid has attracted many researchers in the field of biological fluid dynamics. A bacterial cell contains a single flagellum or multiple flagella attached to different sites of the cell body to propel in a viscous environment. The cytoskeleton of a

flagellum is also known as the axoneme, which is of a flexible rod (elastic in nature)-like structure. The flagellum axoneme undergoes large bending deformations under the action of Dynein motor proteins that generate trust for propulsion in a viscous fluid [1]. The interaction of the active dynein links with passive structural elements (nexin links and radial spokes) produces a coordinated propagation of bending waves along the flagellum length. The exact mechanism by which a flagellum produces its motion has not yet been elucidated until now. Understanding the interactions between the elastic structure of a filament and viscous fluid is a crucial step in developing microfluidic systems powered by artificial swimmers and cilia.

Inspired by bacterial flagella motion in a viscous environment, during the past few years, many theoretical and numerical studies [2–7] have been carried out to elucidate the rotational dynamics of the elastic rod at a very low Reynolds number, Re. In the conventional computational approaches such as the arbitrary Lagrangian–Euler method (ALE) [8–10], a body-fitted grid is used to simulate fluid–structure interaction (FSI) problems involving complex boundaries. These kinds of conformal mesh methods require frequent mesh generation and reconstruction techniques, which require a complex algorithm and are computationally intensive to simulate FSI problems of elastic bodies. Furthermore, a large deformation of elastic bodies may lead to grid quality deterioration. Conversely, in a non-conformal mesh method, a fixed Cartesian grid is used, which is simpler to handle FSI problems. The immersed boundary method (IBM) developed by Peskin [11,12] has been successfully used by several researchers to solve fluid flow and heat transfer problems with complex moving boundaries on a fixed Cartesian grid system.

Camalet et al. [2] considered a two-dimensional model for the beating motion of an elastic filament in a viscous fluid and showed that the wave patterns induced by the filament motion were only dependent on the filament bending rigidity and the viscous drag. They restricted their study to the linear regime of instability and small deformations. Wolgemuth et al. [3] extended the Camalet et al. [2] study to a three-dimensional rotational motion of an elastic rod subjected to a twisting force at one of its ends by considering nonlinear instabilities. They observed two stages for the rod's rotational motion: twirling (the rod rotates about its centerline) and whirling (the rod's centerline wriggles and crankshafts around its rotational axis in a steady state). They also obtained a value for critical frequency, where there is a shape transition from twirling to whirling. Lim and Peskin [4] investigated the same problem using the immersed boundary method by considering a neutrally buoyant elastic filament rotational motion that mimics a flagellar moment in a viscous fluid. They modeled the rod with a flexible cylindrical tube and assigned a set of immersed boundary (IB) points at the inner and outer layer of the cylinder, and the fluid flow was solved by the Navier–Stokes equations. They reported that their filament also undergoes an over-whirling (a subcritical shape transition from whirling motion, where the filament almost folds back on itself) motion in addition to twirling and whirling motions. With the flexible cylindrical tube model, they could obtain the critical frequency value, which was about 3.5 times smaller than that reported in Ref. [3].

Wada and Netz [5] investigated the nonlinear dynamics of an elastic filament subjected to a rotational force at its base with the Langevin dynamics to study the effect of thermal fluctuations on the whirling behavior of the filament. They found that thermal fluctuations play a crucial role in the transition behavior of the filament from whirling to over-whirling. Manghi et al. [6] used the Stokesian simulations technique to study the dynamics of an elastic nanorod rotational motion. They observed that, at a critical torque, the nanorod that was initially held straight and slightly tilted undergoes a shape bifurcation to a helical state. In a review paper, Powers [7] provided an expression for the critical frequency of the twirling and whirling motion of an elastic rod immersed in a viscous fluid by using the slender-body and resistive-force theories. Their expression can roughly estimate the critical frequency obtained in Ref. [4]. Maniyeri et al. [13] simplified the Lim and Peskin [4] method by assigning IB points only at the outer surface of the flexible cylinder immersed in a viscous fluid. They reported that with their simplified model, they could reproduce the same

simulation results of Lim and Peskin qualitatively with very few computational resources. Goldstein et al. [14] studied the dynamics of a twisted elastic filament in a viscous fluid using a nonlinear theory that dynamically couples the twist and bend degrees of freedom. They observed that the twisted filament relaxed into a straight shape (geometric untwisting) only by bending elasticity rather than by axial rotation. Coq et al. [15] investigated the propulsive dynamics of a flexible filament in a viscous fluid with a simple linear model and observed that the propulsive force rose monotonically with the torque amplitude when they applied two transverse oscillating torques, and a discontinuous shape transition occurred when the filament was subjected to a constant axial torque. They validated their theoretical results with an experiment and found that the filament oscillating dynamics strongly depend on the anchoring conditions. Qian et al. [16] studied the deformation of an elastic rod rotating in a viscous fluid numerically and experimentally. They applied a constant torque at the rod base and found that, at low torque, the rod bends gently and creates a small propulsive force, and, at a critical torque, the rod undergoes a helical shape with increased propulsive force.

The lattice Boltzmann method (LBM) [17,18] has emerged as a prominent computational tool for solving various complex fluid flow problems during the past several decades. In LBM, one solves for the particle density distribution functions (PDF) (by obtaining the solution for the Boltzmann kinetic equation on a discrete lattice mesh) instead of directly solving the pressure and velocity fields. The pressure and velocity fields can then be obtained by evaluating the hydrodynamic moments of PDF [18]. Because of its several advantages [19] compared to the conventional Navier–Stokes equations solvers, in recent decades, LBM has been widely used as an alternative CFD tool to conduct mathematical analysis on various multiphysics problems [20–25]. Research on developing a numerical technique based on coupling LBM and IBM for solving fluid–structure interaction (FSI) problems has attained considerable attention among the CFD community to utilize the features of both LBM and IBM. Simulation of biologically inspired fluid dynamic problems by using immersed boundary lattice Boltzmann method (IBLBM) has been a research hotspot in recent years. Some of the research studies are transient deformation of elastic capsules in simple shear flow at low and moderate Re [26], migration and aggregation of red blood cells (RBC) in a two-dimensional micro-channel [27], movement of microparticles in pulmonary acini [28], deformation behavior of RBC flowing in a microfluidic device of a rectangular cross-section for different capillary numbers [29], lateral migration of RBC in a circular channel [30], dispersion of nanoparticles in a viscous fluid with RBC Suspension [31], simulation of ion transport through a pentameric ion channel [32] that encoded in COVID-19. From the above literature, we can say that most of the studies based on IBLBM pertained to RBC migration and/or microparticle migration.

As mentioned before, Lim and Peskin [4] and Maniyeri et al. [13] simulated the elastic rod rotational motion using a finite volume method combined with IBM. However, their model consists of a complex network of springs, which does not mimic the real flagella, computationally intensive and difficult to model. The main objective of this work is to develop a simplified finite element model for the flagellum axoneme that exactly mimics the axoneme internal structure that consists of microtubule doublets connected by nexin links and radial spokes. Another objective of this study is to develop a numerical method based on the combination of LBM and IBM methods to utilize the advantages of both methods in dealing with biological fluid systems. The remainder of this paper is organized as follows. The details of the numerical method proposed in this work are presented in Section 2. The results obtained from the present scheme are discussed in Section 3. The concluding remarks of the present work are mentioned in Section 4.

## 2. Simulation Methodology

### 2.1. Simulation Setup

The cytoskeleton of a flagellum is also known as the axoneme, which consists of nine microtubule doublets (a combination of A and B tubules) surrounding a central pair of

singlet microtubules, as shown in Figure 1. Peripheral microtubules are linked to one another via the passive nexin links and active dynein links, and the central microtubules are linked to the peripheral microtubules via the radial spokes. In this work, we developed a three-dimensional model for a flagellum by representing each microtubule doublet of the axoneme as long slender beams that pass through the centers of each doublet. There are a total of eleven microtubules (nine doublets at the periphery of the flagellum and one central pair). To simplify the model and save computational time, we consider six beams at the periphery and one central beam. Figure 2b shows the discretized model of the flagellum. A beam element is used to discretize each doublet (see Figure 2c for details of a single cross-section; the green color elements show beam elements), and the nexin links [black color elements of Figure 2c] and radial spokes [the blue color elements of Figure 2c] are modeled with truss elements.

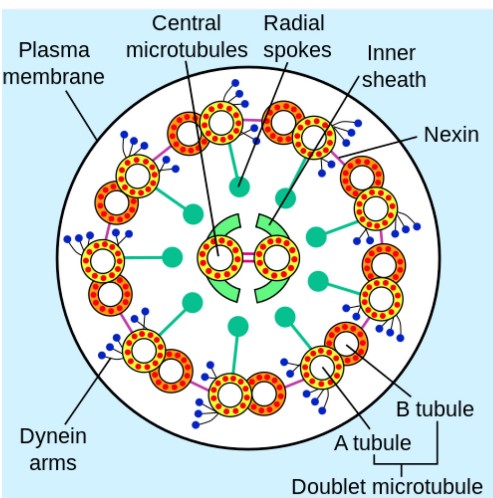

**Figure 1.** Cross-section of an axoneme of a flagellum. Figure is adopted from Wikipedia.

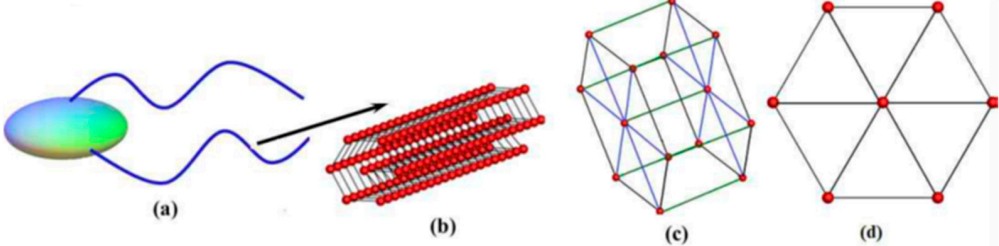

**Figure 2.** (**a**) A bacterial cell with a pair of flagella; (**b**) simulation model of a single flagellum; (**c**) details of each cross-section; (**d**) front view of flagellum when its shape is straight.

Figure 3 shows the simulation setup. All the simulations are performed inside a cubical domain of size 200 μm. The computational domain is split into $41 \times 41 \times 41$ lattice grid points, so that the grid size is $\Delta x = \Delta y = \Delta z \equiv 5$ μm. The periodic boundary conditions are used in the $x$-, $y$-, and $z$-directions. The length and the radius of the elastic rod are set as $L = 100$ μm and $R_a = 5$ μm, respectively. The center of the first cross-section (starting position of the central pair) is located at $[X_C, Y_C, Z_C] = [100, 100, 50]$ μm. The positions for the remaining IB points in the cross-section are calculated based on the equations of the hexagon cross-section. Similarly, the positions for the IB points of other cross-sections are set based on the beam element size of $l_e = 5$ μm. We applied a motor force (discussed in detail in Section 2.2) that acts tangentially to the first cross-section with an angular velocity $w$. The equilibrium configuration of the rod is straight in shape, which is symmetric about the axial axis. Initially, the rod is set in a bent state, i.e., the axial axis of the rod is set slightly inclined with the $z$-axis (rotational axis of the motor). The main objective of this study is to know whether the bent state of the rod attains its equilibrium state (straight state), the bent

state persists, or it increases with the angular frequency of the motor. We used the IBM to simulate the motion of the elastic rod while the fluid flow field was evaluated by the LBM. The following sub-sections will discuss the details of IBM and LBM.

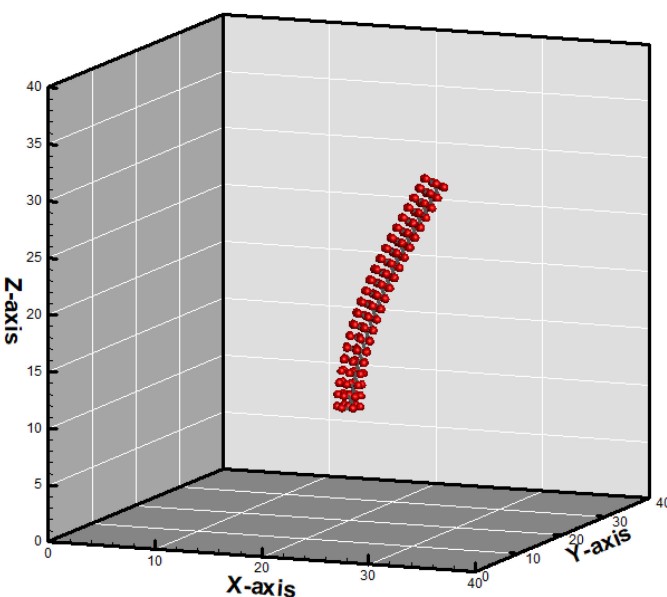

**Figure 3.** The simulation setup employed in the present work.

*2.2. Immersed Boundary Method (IBM)*

The Lagrangian force acting on each node *i* (ranging from 1 to *N*) of the flagellum is obtained by using the following equation:

$$\mathbf{F}_i(\mathbf{X}_i, t) = \mathbf{F}_{\text{ela},i} + \mathbf{F}_{\text{ext},i}, \tag{1}$$

where $\mathbf{F}_{\text{ela},i}$ is the elastic force due to the stretching and bending of a beam element of the flagellum and $\mathbf{F}_{\text{ext},i}$ is the motor force which is modeled by applying a torque on the base (first) cross-section. The elastic force is calculated by

$$
\begin{aligned}
\mathbf{F}_{\text{ela},i} &= -\nabla_i(E_{\text{stretch}} + E_{\text{bend}}) \\
E_{\text{stretch}} &= \frac{1}{2}\sum_{i=1}^{N-1} K_{\text{stretch}}\left(\frac{|\mathbf{X}_{i+1}-\mathbf{X}_i|}{l_e} - 1\right)^2 l_e \\
E_{\text{bend}} &= \frac{1}{2}\sum_{i=2}^{N-1} K_{\text{bend}}\left(\frac{|\mathbf{X}_{i-1}-2\mathbf{X}_i+\mathbf{X}_{i+1}|^2}{l_e^4}\right) l_e
\end{aligned}
\tag{2}
$$

where $K_{\text{stretch}}$, $K_{\text{bend}}$, and $l_e$ in the above equations are the stretching modulus, bending modulus, and the equilibrium length of the element, respectively. The stretching constant value is taken as $K_{\text{stretch}} = 4.5$ μN-m throughout the simulations. The motor force, which is only applied to the first cross-section of the flagella, is modeled by

$$
\mathbf{F}_{\text{ext},i} = \begin{cases} K_{\text{mot}}(\mathbf{X}_{i,\text{tar}}(t) - \mathbf{X}_i(t)) & \text{if } i == 1 \\ 0 & \text{if } i \neq 1 \end{cases}, \tag{3}
$$

where $K_{\text{mot}} = 5$ is motor force constant and $\mathbf{X}_{i,\text{tar}}$ is the target node position at time *t* for achieving a prescribed rotation. $\mathbf{X}_{i,\text{tar}}$ is calculated with the following equation:

$$
\mathbf{X}_{i,\text{tar}}(t) = \begin{cases} X_C + R_a \cos(\theta_i + \omega t), \\ Y_C + R_a \sin(\theta_i + \omega t), \\ Z_C \end{cases}, \tag{4}
$$

After finding $\mathbf{F}_i(\mathbf{X}_i, t)$ from Equation (1), it is spread to the Eulerian grid for calculating force acting by a flagellum node on the fluid using the following equation:

$$\mathbf{f}^{\mathrm{fl}}(\mathbf{x}, t) = \sum_i \mathbf{F}_i(\mathbf{X}_i, t)\delta(\mathbf{x} - \mathbf{X}_i), \tag{5}$$

where $\delta(\mathbf{x})$ is the Dirac delta function used to link the Lagrangian (flagella) and Eulerian (fluid) forces and velocities. The equation for $\delta(x)$ is given by [13] as follows:

$$\delta(\mathbf{x}) = \frac{1}{\Delta x^3}\phi\left(\frac{x}{\Delta x}\right)\phi\left(\frac{y}{\Delta x}\right)\phi\left(\frac{z}{\Delta x}\right), \tag{6}$$

where $(\mathbf{x}) = (x, y, z)$ are the Eulerian coordinates values in the $x$-, $y$-, and $z$-directions. We used the four-point interpolation function for interpolation and extrapolation purposes, which is given by

$$\phi(r) = \begin{cases} \frac{1}{8}\left(3 - 2|r| + \sqrt{1 + 4|r| - 4r^2}\right) & 0 \le |r| \le 1 \\ \frac{1}{8}\left(5 - 2|r| - \sqrt{-7 + 12|r| - 4r^2}\right) & 1 \le |r| \le 2 \\ 0 & 2 \le |r| \end{cases} \cdot \tag{7}$$

After obtaining the force filed on the fluid, the LBE (clear formulation provided in Section 2.3) is used to solve the fluid velocity $\mathbf{u}(\mathbf{x}, t)$ at a Eulerian grid point, $\mathbf{x}$. The velocity at a Lagrangian node (flagellum) is then interpolated with the following equation:

$$\mathbf{U}_i(\mathbf{X}_i, t) = \sum_{\mathbf{x}} \mathbf{u}(\mathbf{x}, t)\delta(\mathbf{x} - \mathbf{X}_i)\Delta x^3. \tag{8}$$

After evaluating the velocity of a flagellum node, the position of the node is updated from the Euler method by using

$$\mathbf{X}_i(t + \Delta t) = \mathbf{X}_i(t) + \mathbf{U}_i(\mathbf{X}_i, t)\Delta t. \tag{9}$$

*2.3. Lattice Boltzmann Equation (LBE)*

In this work, as reported earlier, LBE is used to obtain the fluid velocity field due to flagellum rotation. In LBE, the fluid pressure and the velocity fields are computed from the particle distribution functions, $f_n(\mathbf{x}, t)$, obtained by solving the Boltzmann kinetic equation at a lattice grid point $\mathbf{x}$ at a time $t$ (here, subscript $n$ indicates index for lattice velocity number ranges from 0 to 19 for the D3Q19 lattice). LBE with body force (due to interaction between fluid and flagella) is given by [33,34]

$$f_n(\mathbf{x} + \mathbf{c}_n\Delta t, t + \Delta t) = f_n(\mathbf{x}, t) - \frac{1}{\lambda}\left(f_n(\mathbf{x}, t) - f_n^{eq}(\mathbf{x}, t)\right) + \frac{w_n\Delta t}{c_s^2}\mathbf{f}^{\mathrm{fl}}(\mathbf{x}, t)\cdot\mathbf{c}_n \tag{10}$$

where $\mathbf{c}_n$ is the discrete velocity of $f_n$, $w_n$ is the weighing function of $f_n$, $\lambda$ is the relaxation time, and $c_s$ is the speed of sound. After solving for $f_n$, the fluid density $\rho(\mathbf{x}, t)$, and the velocity $\mathbf{u}(\mathbf{x}, t)$ fields at lattice grid points are obtained from [23,24],

$$\rho(\mathbf{x}, t) = \sum_{n=0}^{b} f_n \qquad \mathbf{u}(\mathbf{x}, t) = \frac{1}{\rho}\sum_{n=0}^{b} f_n\mathbf{c}_n. \tag{11}$$

The fluid density and velocity at the start of the simulation are set as $\rho(\mathbf{x}, t) = 1$ and $\mathbf{u}(\mathbf{x}, t) = 0$, respectively.

**3. Simulation Results**

In this section, we present the simulation results obtained from the simulation of the elastic rod rotational motion in a viscous fluid (for the setup shown in Figure 3) by using the numerical method described in the previous section. We simulated filament rotating motion for three types of motion; twirling motion, whirling, followed by over-whirling motion. In the twirling mode, the rod attains a straight state after some time, i.e., the axial axis (central beam axis) becomes straight, and the entire rod rotates about the axial axis as the bending forces are very high compared to viscous forces. In the whirling mode, the axial axis of the rod is in a bent state, and the tip of the rod rotates about the $z$-axis

(motor rotational axis). The whirling motion is highly unstable, and with a slight increase in the rotational frequency (when the frequency value reaches its peak), the bend of the rod's central axis increases dramatically, and the rod folds back on itself and performs steady-state crank-shafting motion. This kind of motion is known as over-whirling. We qualitatively compared our simulation results with Lim and Peskin's [4] results, as the length scales we have chosen are different from the previous work. As length scales are different, we have chosen the sperm number, $S_p$ (which is the ratio of bending forces and the viscous forces), as a comparing option in such a way that both $S_p$ are identical. The definition of $S_p$ is given by [35]

$$S_p = L \left( \frac{4\pi\mu\omega}{EI} \right)^{1/4}. \tag{12}$$

From Lim and Paskin's work [4], we found that $S_p$ = 6.1 when the angular frequency of motor torque is $f = \frac{\omega}{2\pi} \equiv 1.69$ Hz. We used this value to obtain the bending stiffness $EI$. Table 1 reports the comparison between numerical parameters used in the present work and Lim and Peskin's work.

**Table 1.** Comparison of numerical parameters used in the present work and Lim and Peskin's work.

| Parameter | Present Work | Lim and Peskin [4] |
|---|---|---|
| Flagellum Length, $L$ | 100 μm | 278.2 nm |
| Flagellum Radius, $R_a$ | 5 μm | 11.5 nm |
| Fluid Viscosity, $\mu$ | 0.001 kg/m·s | 0.001 kg/m·s |
| Fluid Density, $\rho$ | 1000 kg/m$^3$ | 1000 kg/m$^3$ |
| Motor Frequency, $f$ | 1.69 Hz | 1.69 Hz |
| Bending Stiffness, $EI$ | $10 \times 10^{-21}$ N·m$^2$ | $6.12 \times 10^{-31}$ N·m$^2$ |
| Reynolds Number, Re | $5.3 \times 10^{-4}$ | $3 \times 10^{-9}$ |
| Sperm Number, $S_p$ | 6.1 | 6.1 |

### 3.1. Twirling Motion

In this section, the simulation results of the elastic rod behavior when it undergoes a twirling motion, where the rod rotates like a rigid membrane about its own axis, are reported. Figure 4 shows the instantaneous three-dimensional shapes of the rod during twirling motion when the angular frequency of the motor is $f$ = 1.5 Hz. The fluid flow surrounding the rod is also clearly shown in Figure 5. The motor torque is applied to the first cross-section, and the rod central beam axis is set at an angle inclined to the motor rotational axis. The first cross-section of the rod undergoes a rotational motion due to applied torque. The rotational motion is then transferred to the successive cross-sections as they are connected with beam elements, which results in the rotational motion for the entire rod. Initially, the rod is in bent form, as shown in Figure 4. However, after some time, the rod attains a straight shape (reaches its equilibrium state), i.e., the axial axis of the rod becomes straight, and the entire rod simply rotates like a rigid body about its axial axis. When the rod rotates, it drags the surrounding fluid so that the fluid surrounding the rod also rotates in the same sense as the rod, as shown in Figure 5. This twirling motion is stable as at lower angular frequencies of the motor; the bending forces are very high compared to the viscous forces.

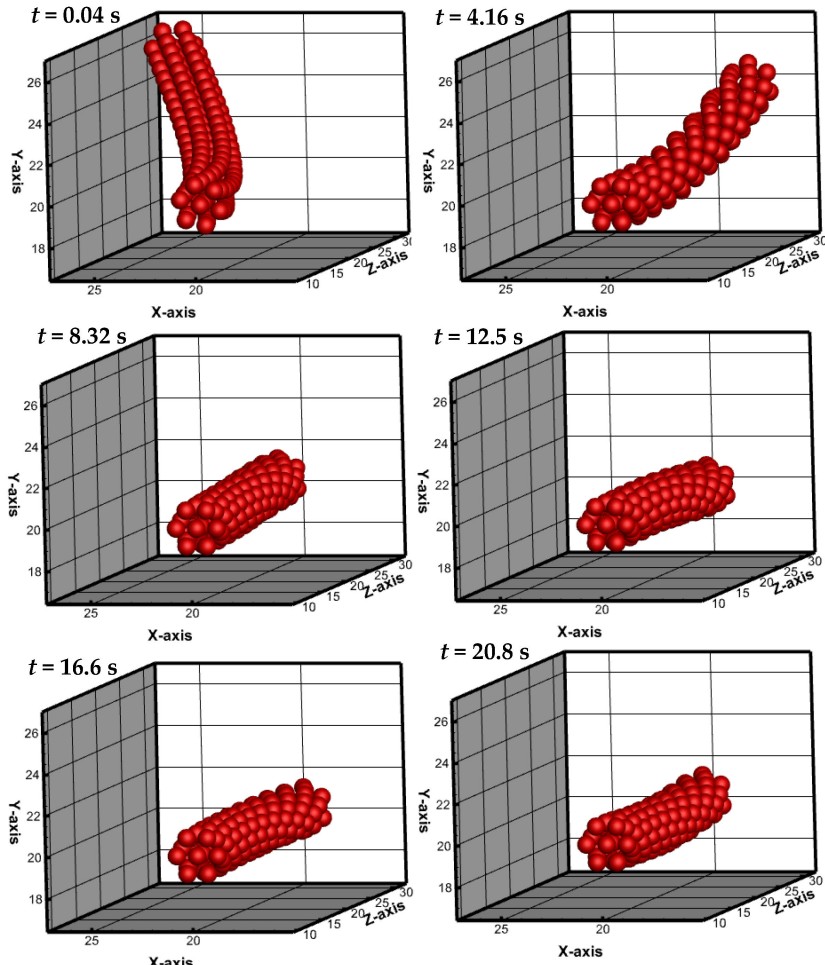

**Figure 4.** The instantaneous shape of the elastic rod when viewing from the motor end during the twirling motion at $f$ = 1.5 Hz.

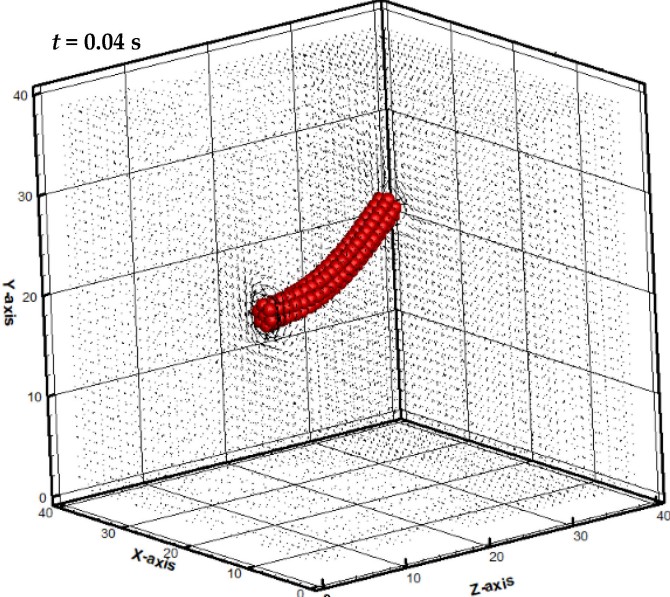

**Figure 5.** *Cont.*

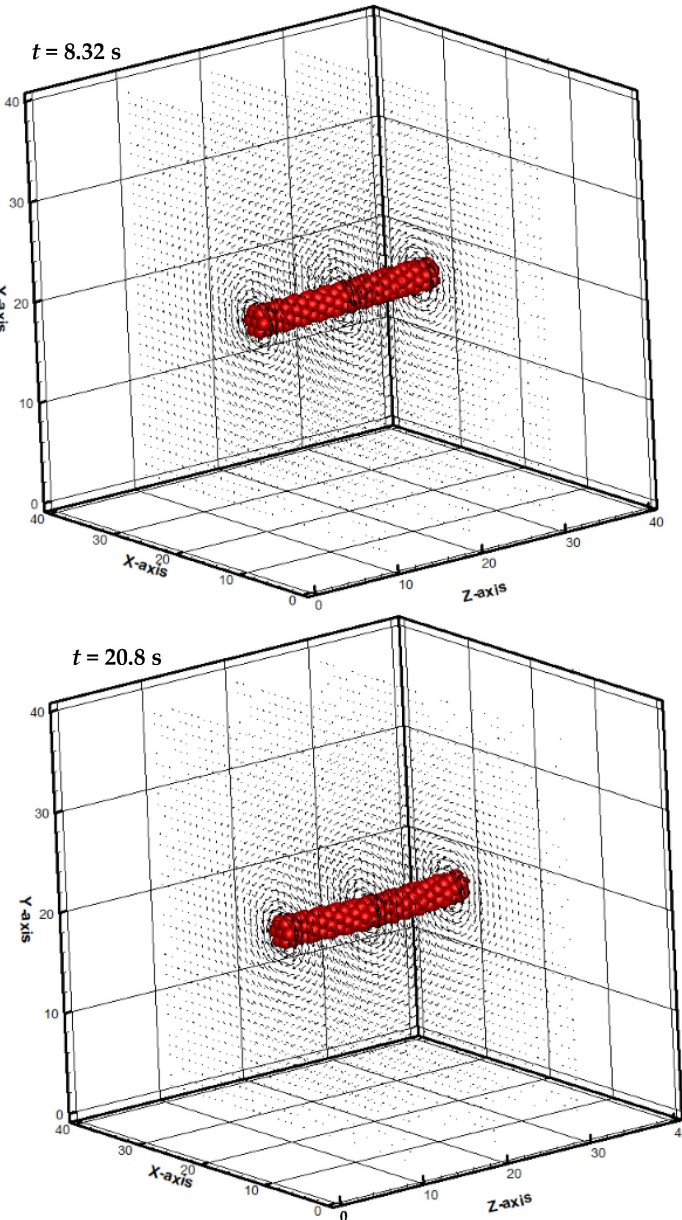

**Figure 5.** The instantaneous flow filed during the twirling motion at $f$ = 1.5 Hz.

### 3.2. Whirling Motion

When the frequency increases beyond a critical value, the twirling motion becomes unstable and is replaced by the whirling motion. In the whirling mode, the rod's axial axis is always in a bent state (does not attain a straight shape), and the entire rod rotates about the symmetrical axis (motor rotational axis, *z*-axis) at an angular frequency that is different from $f$. Therefore, the motion of the filament is not like a rigid body motion. Figure 6 shows the instantaneous three-dimensional shapes of the rod during whirling motion when the angular frequency of motor force is $f$ = 2.65 Hz. From the figure, it is seen that the rod undergoes the whirling motion where the rod initially takes a helical shape, and the rod's free end rotates about the central rotational axis of the first cross-section (the driving motor). Figure 7 shows the instantaneous flow field around the rod during the whirling motion. It is evident from the figures that fluid also rotates with the rod, and the fluid's rotational motion is slightly disturbed at the tip of the rod. This is because the fluid flow rotates along the rod length, and the rod's free end whirls around the central axis of the motor.

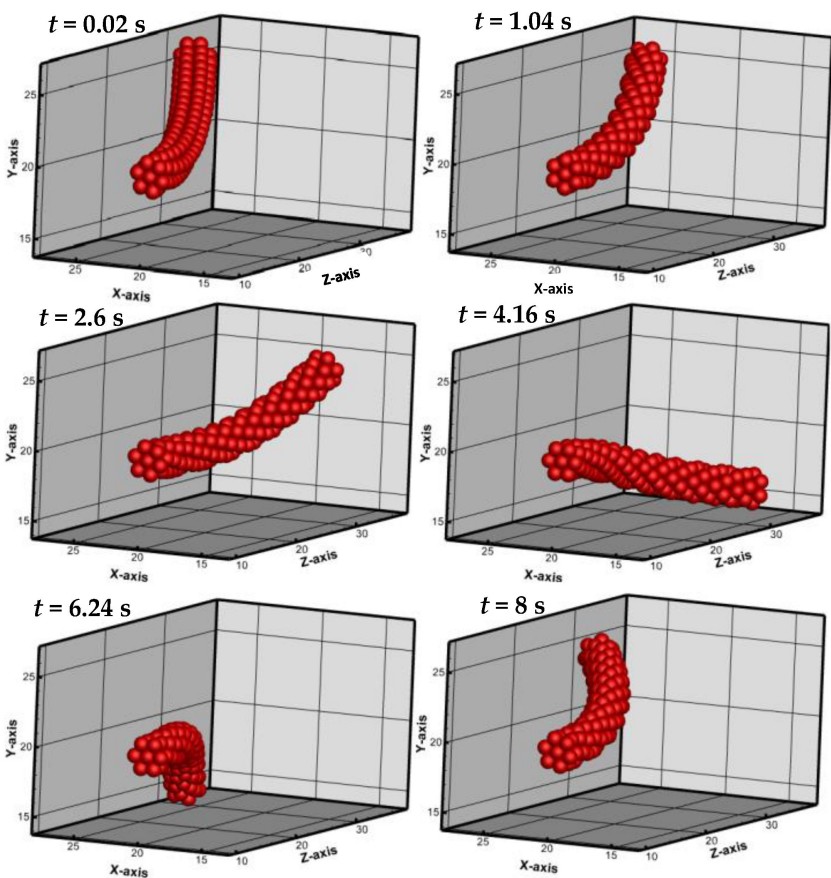

**Figure 6.** Instantaneous shapes of the elastic rod when viewing from the motor end during the whirling motion at *f* = 2.65 Hz.

### 3.3. Over-Whirling Motion

With a slight increase in the frequency (when it reaches its critical value), the bent amplitude of the rod increases, and the whirling motion converts into the over-whirling motion (a discontinuous shape transition occurs) when the angular frequency of the motor is approximately equal to $f_c \approx 2.7$ Hz. Figure 8 shows the instantaneous shapes of the rod during over-whirling motion at *f* = 3.0 Hz. In this mode, the amplitude of the bend of the rod's central axis increases dramatically in such a way that the rod's tip folds back on itself, and after some time, the tip of the rod comes in front of the base as shown in Figure 8 (at *t* = 9.4 s). After folding back, the rod performs a steady crank-shafting motion. Figure 9 shows the instantaneous fluid flow field along with the rod at three different cross-sections during the over-whirling motion. Since the rotational frequency is higher, we can observe a rotating flow field throughout the length of the rod from the beginning of the simulation. As the rod takes a helical shape and folds back on itself, we can observe a pumping action of the fluid in the opposite direction of the *z*-axis as the rod drags the fluid in the reverse direction.

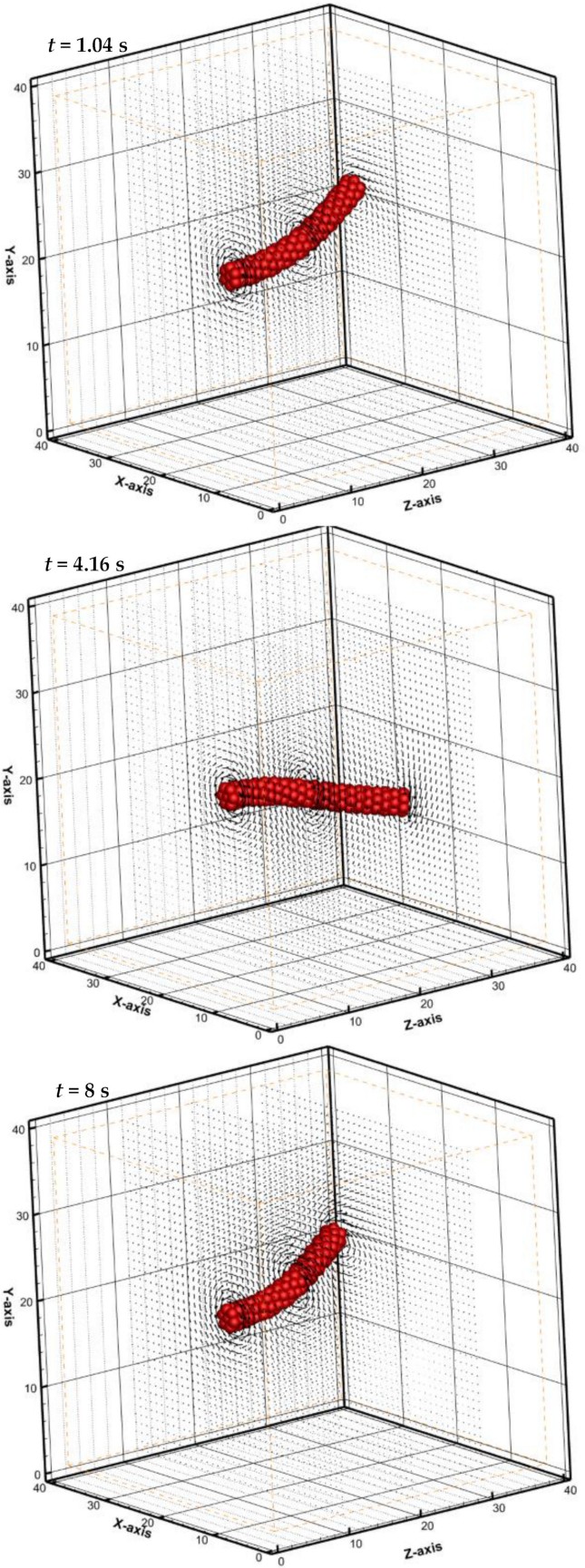

**Figure 7.** The instantaneous flow field during the whirling motion at $f$ = 2.65 Hz.

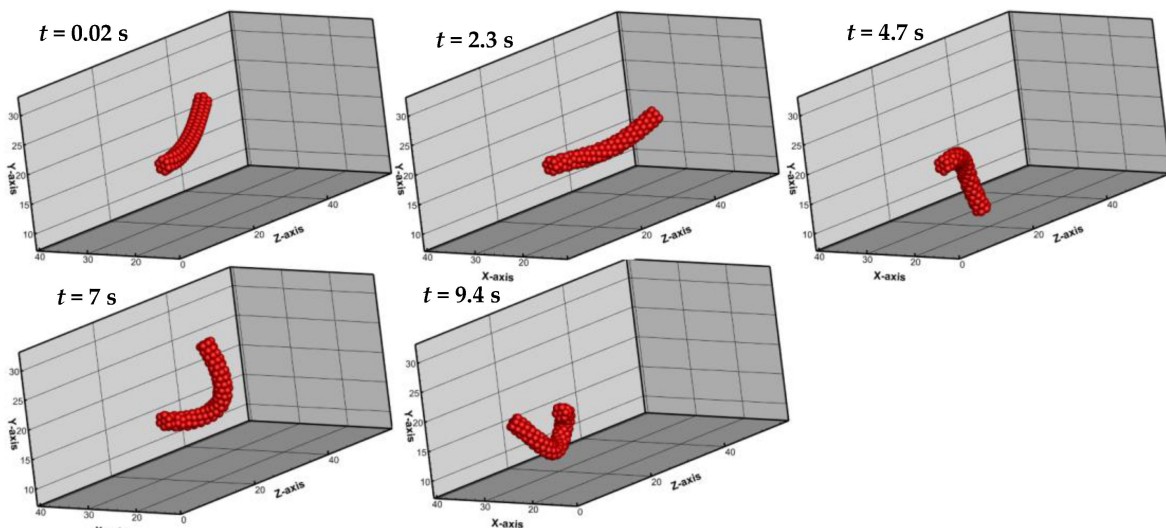

**Figure 8.** Instantaneous shapes of the elastic rod when viewing from the motor end during the over-whirling motion at *f* = 3.0 Hz.

After qualitatively comparing our simulation results with Lim and Peskin's [4] results, we can say that the present simulation results agree well with their results, although we used a different length scale for the flagellum. The critical frequency value from our simulation results also approximately matches Lim and Peskin's results. Lim and Peskin reported that their simulation technique is computationally very intensive (a typical computation requires a total CPU time of 43.75 days). Maniyeri and Kang [13] also used the same length scales as the present work, and they mentioned that their method took a CPU time of 7 days to run a simulation using an Intel® core i7 processor. In this work, by employing a simple FEM model that closely mimics the real flagellum, we could well capture all three rotational dynamics (twirling, whirling, and over-whirling) of the elastic rod rotational motion in a viscous fluid. Our simulation method only requires 1.5 days of CPU time of an Intel® core i7 processor, which means that the present simulation technique is computationally more efficient compared to the methods used in the previous literature. It is worth mentioning that the greater difference in the CPU time between our case and Lim and Peskin's case is mainly due to the difference in the length scales adopted in each case. We can adopt the same length scales as Lim and Peskin's work and can match the value for the critical frequency of shape transition accurately. However, to save computational time, it is not necessary to consider the exact length scales of the real flagellum to elucidate the physics behind the propulsion motion of a bacterial flagellum, as the Reynolds number is very low at both the length scales, Re $\ll$ 1. Even if we used the same length scales, we believe that the CPU time is much less than that of Lim and Peskin's work due to the simple FEM model of our scheme. We adopted the same length scales as Maniyeri and Kang's work and observed that the CPU time of our work is 4.5 times smaller than that of Maniyeri and Kang's. Our simulation model can be straightforwardly extended to solve the motion of an artificial bacteria used in various microfluidic systems.

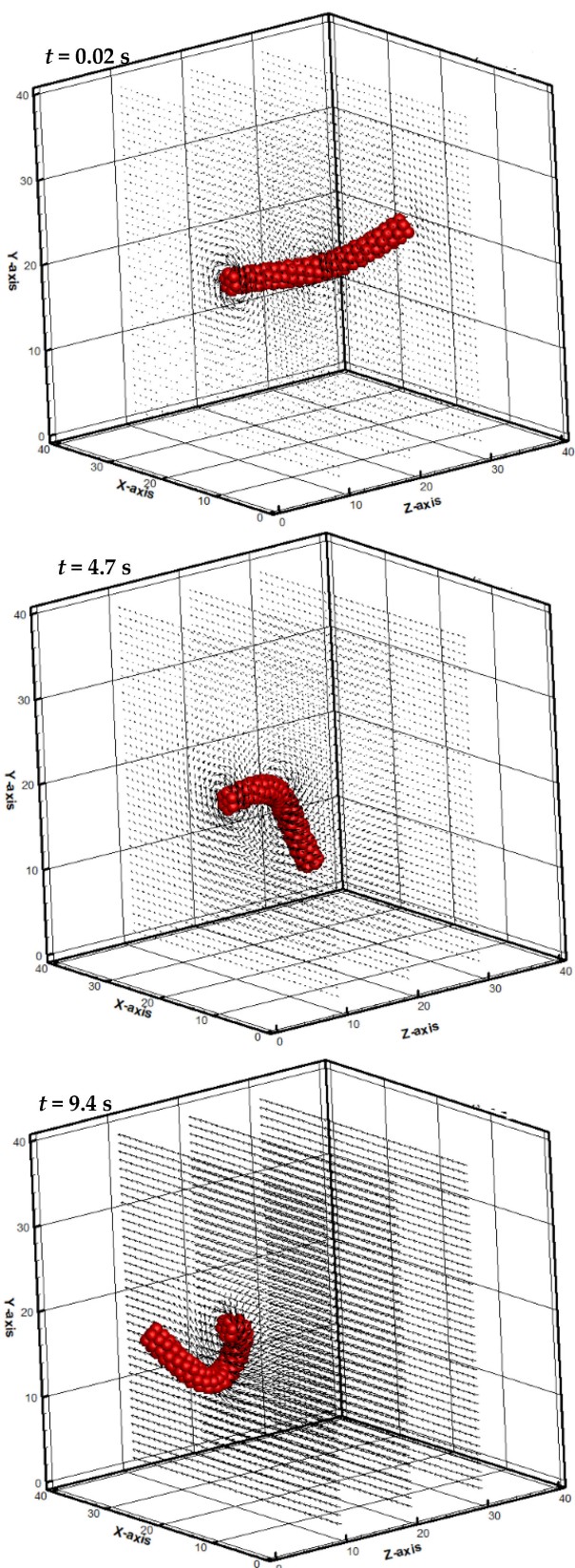

**Figure 9.** The instantaneous shape of the elastic rod during the over-whirling motion when *f* = 3.0 Hz.

## 4. Conclusions

In this work, we developed a three-dimensional finite element model to simulate the rotational motion of an elastic rod that mimics a flagellum axoneme in a viscous fluid by

using the immersed boundary lattice Boltzmann method. The flagellum's microtubules arrangement, and nexin links and radial spokes are modelled with beam and truss elements, respectively. The elastic force on each node of the flagella is obtained from the variational derivative of total stretching and bending potentials. The fluid flow field is solved by LBM, while the hydrodynamic interactions between fluid and flagellum are treated with IBM. A rotational torque is applied at the base of the rod, and simulations are performed by varying frequency of rotation in the range of 1.5 Hz to 3.0 Hz. With the simplified model (by avoiding the complex network of the spring used in the previous literature), we could simulate all three regimes of rotational motion, viz. twirling, whirling, and over whirling reported in the previous literature. We could also match the critical frequency value, where whirling motion transits into over-whirling motion, obtained from the present simulation with Lim and Peskin's results. After comparing the CPU time, it is found that our simulation model is computationally more efficient compared to the methods proposed in the previous literature.

**Author Contributions:** Conceptualization, S.A. and W.C.; methodology, S.A., W.C., S.S.R. and G.T.T.P.; software, S.A., W.C., S.S.R. and G.T.T.P.; formal analysis, S.A., W.C. and G.T.T.P.; investigation, S.A., W.C., S.S.R. and G.T.T.P.; writing—original draft preparation, S.A. and G.T.T.P.; writing—S.A., W.C., S.S.R. and G.T.T.P.; visualization, S.A., W.C., S.S.R. and G.T.T.P.; supervision, S.A. and G.T.T.P.; project administration, S.A. and W.C.; funding acquisition, S.A. and W.C. All authors have read and agreed to the published version of the manuscript.

**Funding:** This research was supported by the National Research Foundation of Korea (NRF) grant funded by the Korean government (Ministry of Science and ICT) (2020R1G1A1010247).

**Data Availability Statement:** Not applicable.

**Acknowledgments:** The authors acknowledge the support of Yann Ling Yang, currently a student in the Mechatronics Engineering department of Kyungsung University, for executing some of the simulations.

**Conflicts of Interest:** The authors declare no conflict of interest.

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
