# Peer review of "Simulation of an Elastic Rod Whirling Instabilities by Using the Lattice Boltzmann Method Combined with an Immersed Boundary Method"

_axioms, doi:10.3390/axioms12111011_

Round 1

Reviewer 1 Report

Comments and Suggestions for Authors

Review comments are attached.

Author Response

Dear reviewer,

We thank you for reviewing our manuscript and providing valuable comments/suggestions to improve our paper. Below, we provide the detailed responses to your comments.

  1. Comment #1: I feel that the authors mentioned only a few pieces of literature related to elastic rod rotational motion in a viscous fluid. I recommend adding more references in the introduction section

 <Authors Response>

As per the reviewer’s recommendation, we added some references [13, 14, 15] related to theoretical and experimental studies on elastic rod rotational motion in a viscous fluid to our new version of the manuscript (one can refer to lines 98~111 of page #3)

  1. Comment #2: In the simulation methodology section (Figure 2(b)), why did the authors only consider six microtubules for the flagellum instead of nine (most of the flagella consist of nine doublets at the periphery)? Please explain

 <Authors Response>

We agree with the reviewers’ comment that most of the flagella consist of nine microtubules doublets at the periphery. However, to save computational time (without losing accuracy), we considered six microtubules for the flagellum modelling. This procedure was adopted in many previous literatures. We already mentioned this point in our previous version of the manuscript (one can refer to lines 159~162 of page #4)

  1. Comment #3: There are no values for Kstretch and Kmot in the present version of the manuscript

 <Authors Response>

We added the values for Kstretch and Kmot  to our new version of the manuscript (one can refer to lines 208~209 of page #6 and line #212 of page #6)

  1. Comment #4: What kind of Dirac delta function is used for interpolating velocities and forces? If possible, provide an equation for it.

<Authors Response>

We used a four-point interpolation formula for the Dirac delta function. We also added the equation [Equations ()6 and (7)] for the Dirac delta function to the new version (one can refer to lines 219~224 of page #6)

  1. Comment #5: Provide a reference for Sperm number Sp [Equation (10)].

<Authors Response>

We added the reference for Sperm number Sp [Equation (12) of modified version] (one can refer to line 271 of page #7)

  1. Comment #6: In Eqns. (5) and (6) why negative sign is used

<Authors Response>

We removed the negative sign in front of Equation (5) and Equation (6)

Reviewer 2 Report

Comments and Suggestions for Authors

The manuscript is good and the results are presented in a nice manner. However, I would like to see a larger discussion about the difference in length scales of the problem, when comparing it to Lim and Peskin.

The discussion provided is very short and the reader is not sure why such a difference was chosen. Is it possible to simulate the exact conditions as Lim and Peskin?

They need to explain and discuss why they are simulating a problem with a totally different length scale from the problem they are using for validation. They also need to explain if it is possible to simulate the exact conditions of those from Lim and Peskin with their simplified method. The reader is unsure if the gain in computational speed is due to the smaller scale of the problem or not.

Comments on the Quality of English Language

Can be improved

Author Response

We partially agree with the reviewer's comment that one reason for the gain in computational speed in our work is we adopted larger length scales compared to Lim and Peskin's work [4]. In this work, we have chosen the same length scales adopted in Maniyeri and Kang's work [12].  We can adopt the same length scales as Lim and Peskin's work and can match the value for the critical frequency of shape transition accurately. However, to save computational time, it is not necessary to consider the exact length scales of the real flagellum to elucidate the physics behind the propulsion motion of a bacterial flagellum as the Reynolds number is very low at both the length scales,  . Even if we used the same length scales, we believe that the CPU time is much less than that of the Lim and Peskin work due to the simple FEM model of our scheme. When we compare the CPU time of the present work with Maniyeri and Kang's work it is observed that the CPU time of our work is 4.5 times smaller. We added this discussion to our modified version of the manuscript 
